# Diversity of nutritional content in seeds of Brazilian common bean germplasm

Jessica Delfini[1,2], Vânia Moda-Cirino[2], José dos Santos Neto[2], Juliana Sawada Buratto[2], Paulo Maurício Ruas[3], Leandro Simões Azeredo Gonçalves[1]*

1 Department of Agronomy, Universidade Estadual de Londrina (UEL), Londrina, Paraná, Brazil, 2 Instituto de Desenvolvimento Rural do Paraná –IAPAR–EMATER (IDR–IAPAR–EMATER), Londrina, Paraná, Brazil, 3 Department of Biology, Universidade Estadual de Londrina (UEL), Londrina, Paraná, Brazil

* leandrosag@uel.br

**Data Availability Statement:** All relevant data are within the paper and its Supporting Information files.

## Abstract

Mineral deficiency is worldwide one of the major problems associated with human health, and biofortification through breeding is considered an important strategy to improve the nutritional content of staple food in countries that face this problem. The assessment of genetic variability for seed nutrient contents is a first step in the development of a biofortified crop. From the germplasm bank IDR–IAPAR–EMATER, a set of 1,512 common bean accessions, consisting of local and commercial varieties and improved lines, was analyzed. High variability among the accessions was observed for all evaluated nutrient contents (P, K, Ca, Mg, Cu, Zn, Mn, Fe and S and protein). In the mean, the contents of the carioca and black market groups (Mesoamerican gene pool), were around 7% higher for the minerals Ca, Cu, Mn and Fe and between 2–4% higher for P, K, Mg and Zn than in the other groups with Mesoamerican and Andean common bean. Few differences were observed among the Mesoamerican accessions that belong to the carioca and black commercial groups. Wide variability was observed among the evaluated genotypes, and the concentrations of the best accessions exceeded the overall mean by 14–28%. Due to the high variability in the evaluated accessions, these results may contribute to the selection of promising parents for the establishment of mating blocks. The nutritional contents of many of the improved lines evaluated in this study were higher than those of the commercial cultivars, indicating the possibility of developing new biofortified cultivars.

## Introduction

Minerals are inorganic substances, present in all tissues and fluids of the human body, which play an important role in growth and regulation processes and repair functions of the body [1]. However, more than 2 billion people worldwide have one or more chronic micronutrient deficiencies, particularly of calcium (Ca), iodine (I), iron (Fe), selenium (Se), zinc (Zn) and vitamins [2]. Micronutrient deficiency can delay the growth and cognitive development, impair immune functions and increase the risk of cardiovascular and metabolic diseases [3].

Several strategies, e.g., of supplementation (e.g., in the form of pills), industrial fortification and biofortification, have been applied to combat micronutrient malnutrition.

**Funding:** The funders had no role in study design, data collection and analysis, decision to publish, or preparation of the manuscript.

**Competing interests:** he authors have declared that no competing interests exist.

Supplementation and industrial fortification have some practical limitations, since the supplementation pills are not always taken regularly, while industrial food fortification requires a rigorous quality control, with specialized techniques and infrastructures, which are generally restricted in developing countries [4,5]. On the other hand, biofortification is considered a viable and economical means of providing populations who have limited access to dietary diversification and other micronutrient interventions with essential micronutrients [6,7].

Biofortification can be defined as the process of raising the concentration of vitamins and minerals in a crop by plant breeding (conventional methods or genetic engineering) or by agronomic practices (agronomic biofortification). Agronomic biofortification is the process of improving the micronutrient content in edible plant parts by applying fertilizers to the soil or leaves. However in the long term, genetic biofortification is more cost-effective [8].

Several studies have demonstrated the effectiveness of conventional breeding in developing biofortified crops, resulting in an improved micronutrient intake and minimizing deficiency among the target populations. Some examples are higher vitamin A contents in sweet potato [9,10] maize [11] and cassava [12], as well as increased Fe contents in common bean [13,14] and millet [15,16]. Altogether, more than 150 biofortified varieties of 10 different crops have been officially released for production in over 30 countries in Africa, Asia and Latin America and the Caribbean [7].

In conventional breeding, several steps are indispensable to establish biofortified cultivars. The first is to analyze the variability contained in germplasm banks in order to estimate the genetic potential of some accessions for future stages of the breeding program [17]. In an evaluation of the common bean (*Phaseolus vulgaris* L.) core collection of the International Center for Tropical Agriculture (CIAT), Islam et al. [18] found wide genetic variability for nutritional traits (mean Fe and Zn contents of 54.81 ppm (34.6–91.9 ppm) and 34.40 ppm (20.7–59.4 ppm) respectively). In a collection of 1,150 accessions, wide variability for mineral contents was also reported by Beebe, Gonzalez and Reginfo [19], who concluded that the Fe and Zn grain contents could be increased. The mean Fe and Zn values reported by these authors were 55 ppm (34–89 ppm) and 35 ppm (21 to 54 ppm), respectively. These studies showed evidences that the Mesoamerican gene pool have a tendency to have higher contents of calcium, phosphorus, sulfur and zinc than the Andean gene pool, but in the other hand, Andean accessions showed higher iron contents.

The common bean breeding program of the Rural Development Institute of Paraná – IAPAR–EMATER (IDR–IAPAR–EMATER) currently has a collection of 14,163 accessions deposited by different research institutions and universities, as well as lines developed by the institute itself [20]. These accessions are being characterized in detail and some of them extensively exploited in the breeding program for the development of new cultivars. To date, 39 common bean cultivars have been released, which are being widely cultivated by producers across Brazil. In this sense, the development of new cultivars that combine high yields with resistance or tolerance to biotic and abiotic stresses and high nutritional value becomes increasingly relevant. Therefore, this study evaluated the diversity of the mineral composition and protein content of common bean accessions of the IDR–IAPAR–EMATER germplasm bank, with a view to identifying accessions that can be used as parents in breeding programs to develop cultivars with seeds with higher nutritional value.

## Material and methods

### Plant material and experimental design

Of the germplasm bank of the IDR–IAPAR–EMATER, 1,512 accessions were evaluated. All information about seed coat color, genetic material and center of origin (1,473 and 39

belonging to Mesoamerican and Andean gene pool, respectively) of the accessions are described in the S1 Table.

The experiments were carried out at different locations in the state of Paraná, Brazil, in two agricultural seasons, resulting in a total of four environments. In the rainy season of 2009, sowing was carried out in September in the counties of Pato Branco (lat. 260˚ 13'S; long. 520˚ 40'W 760 m asl,) and Lapa (lat. 250˚ 46 'S; long. 490˚ 42'W; 908 m), and in the dry season of 2010 in January, in Ponta Grossa (250˚ 05'S; long. 500˚ 09'W; 975 m asl) and Lapa.

Each experimental plot consisted of one 4-m row, at a spacing of 0.50 m between rows and a density of 12 plants per meter. Fertilization at sowing consisted of the application of 300 kg ha$^{-1}$ of 4-30-10 (N, $P_2O_5$, $K_2O$) and in growth stage V4, of 200 kg ha$^{-1}$ ammonium sulfate. Pests, diseases and weeds were controlled according to the technical recommendations for the crop. At physiological maturity (R9), the plots were harvested and a 100-g sample of disease-free seeds without physical or insect damage was collected and stored in a cold chamber (5.6˚C, humidity 33%) until laboratory analysis.

## Analysis of mineral composition and protein content

The following minerals were determined: phosphorus (P), potassium (K), calcium (Ca), magnesium (Mg), copper (Cu), zinc (Zn), manganese (Mn), iron (Fe), sulfur (S) and nitrogen (N). Prior to laboratory analyses, the samples were washed in tap water, 0.01M HCl solution and distilled water to prevent contamination by soil particles that could be attached to the seeds. Then the seeds were oven-dried at 60˚C for 48h, ground and the flour stored in a glass recipient.

The mineral contents were determined by nitroperchloric digestion with $HNO_3$:$HClO_4$ solution and by atomic emission spectrophotometry (ICP-AES) (Thermo Jarrell Ash ICAP 61E). The protein content was quantified by Kjeldahl's method and the samples were read on a UV-VIS spectrophotometer. Factor 6.25 was used to convert total seed nitrogen into crude protein [21]. The analysis was developed according to the methodology described by Miyazawa et al. [22].

## Statistical analysis

The data were subjected to descriptive analysis (minimum, maximum and mean values; coefficient of variation; asymmetry and kurtosis), using software SAS [23]. Pearson's correlation analysis using the mean values for each accession was performed using the cor() function in the R package qgraph [24]. The accessions were separated based on two criteria: seed coat color (black, carioca and colored) and genetic material (cultivars, landraces and breeding lines). For these two criteria an analysis of variance (ANOVA) followed by a Tukey test ($p < 0.05$) were performed for each case to compare significantly differences between the formed groups. To visualize the distribution of the accessions grouped by each criteria a boxplot graph was made. The analyses were done using the R packages agricolae and ggplot2 [25–27].

Subsequently, multivariate analysis using the Ward-Modified Location Model (Ward-MLM) method was performed to group the accessions by the procedures CLUSTER and IML of software SAS [23]. For the Ward clustering method, the distance matrix was calculated by the Gower algorithm [28]. The ideal number of groups was defined according to the criteria pseudo-F and pseudo-$T^2$, together with the likelihood profile associated with the likelihood ratio test. The data distribution in relation to the groups formed was visualized in a boxplot.

The differences between groups and correlation of the variables with the canonical variable were represented in a diagram using the SAS CANDISC procedure [23]. The distance proposed by Matusita [29], used by Krzanowski [30] and later by Franco et al. [31] for the distribution of variables, was applied to determine the dissimilarity between the groups.

## Results

Wide variability for mineral and protein contents was detected in the accessions of the IDR–IAPAR–EMATER common bean germplasm bank (Table 1). The coefficients of variation (CV) ranged from 5.66 (protein) to 13.53% (Ca), indicating good experimental precision. The kurtosis values, which indicate the degree of flatness of the data distribution, were close to 3.0 for protein, Zn and P, suggesting a mesokurtic distribution. The other nutrients were classified as leptokurtic, in which the data distribution curve is close to the central value that is greater than the mesokurtic distribution. The asymmetry was positive for Ca, Zn, Mn, Fe and protein, and negative for the other nutrients. Most nutrient distributions were not normal (all except P, Fe and protein) (Table 1).

Positive correlations were observed among most nutrients. Protein content correlated only with P, while K correlated with P and Zn. In addition, two correlation groups (Fe-Zn-Cu-P and Ca-Mn-Mg) were formed, in which all nutrients involved are correlated with each other (Fig 1).

In the subdivision of accessions by seed coat color (Fig 2), the means of the colored group accessions, were lower than those of the carioca and black groups, except in the case of protein and S contents. The means of the colored group were around 7% lower for the minerals Ca, Cu, Mn and Fe and between 2–4% lower for P, K, Mg and Zn. Significant differences between the mean contents of the black and carioca groups were only observed for K, Mn and S; the black group had highest K and S and the carioca group highest Mn contents. However, some accessions of the black group also had high Fe and some accessions of the carioca group high Zn contents (Fig 2).

The variability in the breeding lines was higher than in the cultivars and landraces, aside from the higher mean contents for all nutrients except protein. The mean P, K and Zn concentrations of the breeding lines, cultivars and landraces were similar, while the breeding line means were significantly higher for Ca, Mg, Cu, Mn, Fe and S and lower for protein. For Fe content, the breeding line mean was 5% higher than that of the other groups (Fig 2).

By the logarithmic likelihood function, five groups were formed based on the criteria pseudo-$t^2$ and pseudo-F (S1 Fig and S1 Table). In the boxplot representing the groups, variability was observed between and within them (Fig 3 and Table 2).

**Table 1. Descriptive statistics for mean mineral and protein contents evaluated in common bean samples of 1,512 accessions of the germplasm bank of the Rural Development Institute of Paraná –IAPAR–EMATER (IDR–IAPAR–EMATER), grown in four environments in the state of Paraná, Brazil.**

| Nutrient | Minimum | Mean | Maximum | CV[a] | Asymmetry | Kurtosis | Normal Distribution (Yes/No)[b] |
|---|---|---|---|---|---|---|---|
| P (g.kg$^{-1}$) | 4.66 | 6.52 | 8.38 | 8.36 | -0.11 | 3.32 | Yes |
| K (g.kg$^{-1}$) | 12.25 | 17.37 | 21.75 | 6.13 | -0.28 | 4.20 | No |
| Ca (g.kg$^{-1}$) | 0.86 | 1.49 | 2.40 | 13.53 | 0.13 | 3.58 | No |
| Mg (g.kg$^{-1}$) | 1.71 | 2.22 | 2.66 | 5.88 | -0.40 | 4.07 | No |
| Cu (mg.kg$^{-1}$) | 7.83 | 12.55 | 17.63 | 9.77 | -0.13 | 4.04 | No |
| Zn (mg.kg$^{-1}$) | 26.45 | 36.15 | 46.90 | 7.80 | 0.27 | 3.20 | No |
| Mn (mg.kg$^{-1}$) | 11.50 | 16.59 | 23.32 | 10.33 | 0.49 | 3.77 | No |
| Fe (mg.kg$^{-1}$) | 51.84 | 80.48 | 115.27 | 9.94 | 0.16 | 3.51 | Yes |
| S (g.kg$^{-1}$) | 1.71 | 2.57 | 3.09 | 8.04 | -0.85 | 3.83 | No |
| Protein (g.kg$^{-1}$) | 169.31 | 202.14 | 248.43 | 5.66 | 0.18 | 3.06 | Yes |

[a] CV = Coefficient of variation;

[b] Kolmogorov-Smirnov Normality Test.

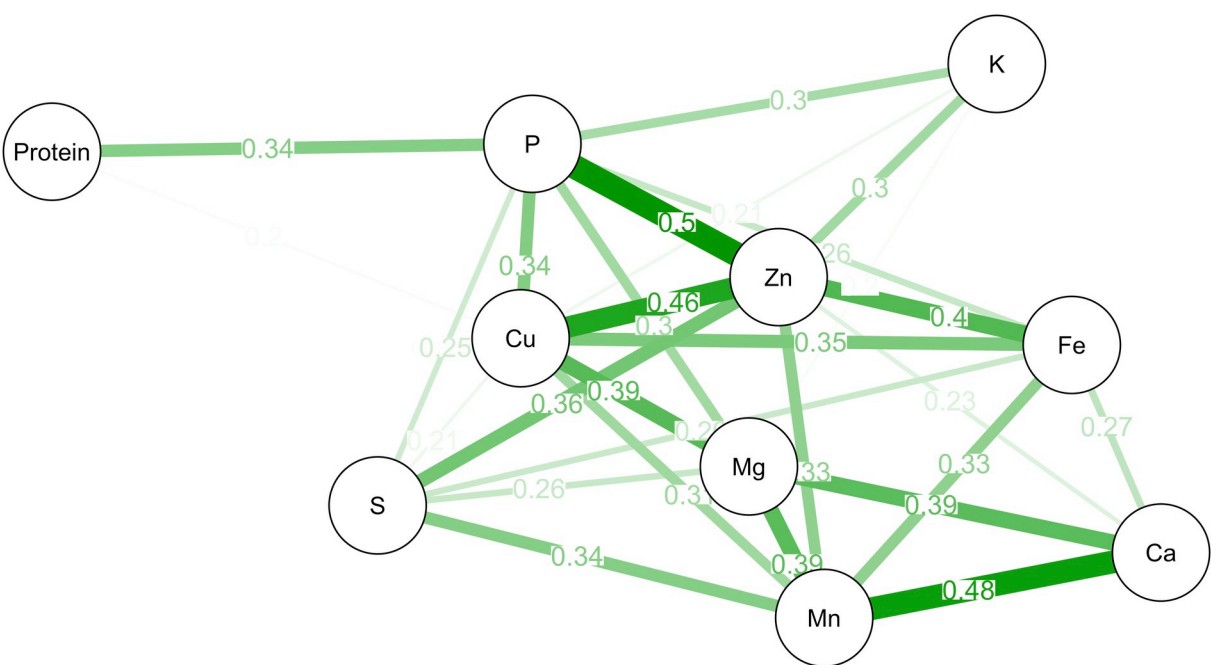

**Fig 1. Graphical representation of Pearson's correlation among the grain nutrient contents of 1,512 common bean accessions of the germplasm bank of the Rural Development Institute of Paraná –IAPAR–EMATER (IDR–IAPAR–EMATER).** Only edges of significant correlations are shown in the graph ($p \leq 0.05$).

Group 1 (G1) consisted of only 80 accessions, mostly with colored grain, and included most Andean accessions of this study, which accounted for approximately 31% of all accessions of the group. This group had the lowest contents for 8 of the 10 evaluated nutrients. Group 2 (G2) was the largest, with 52.2% of the studied accessions, with mainly black and carioca grain (71%). In general, the variability in G2 was the highest while the means were not notable for any of the nutrients. However, some accessions had high Ca, Mg and Fe contents. Group 3 (G3) comprised 202 accessions, of which 122 had black, 44 mulatto and the remaining accessions other seed coat colors. Group G3 stood out with the highest P, Cu, Zn and Fe, and some accessions had very high K contents.

Group 4 (G4) comprised mostly accessions of the commercial groups carioca and black, as well as brown, mulatto and others. Mainly the Mn content was very high in G4. Together with G3, the S content was also very high in G4. In the fifth group (G5), only carioca accessions were included, the S content was the lowest and the Mn and Fe contents were rather low, whereas the mean protein contents were the highest.

The first two canonical variables explained 67.64 and 19.44%, respectively, resulting in 87.07% of the total variation (Fig 4). The nutritional contents that contributed most to determine the genetic diversity among the accessions studied in the first canonical variable were S and Mn, while the most important for the second canonical variable were Zn, P and Cu (Table 2). The genotype dispersion based on the first two canonical variables is shown in Fig 4.

The distances between groups by the Ward-MLM strategy based on the distance proposed by Matusita [29] indicated that the shortest distance was between G2 and G3 (estimated value of 7.23). This indicates high similarity between the accessions of these groups. The highest dissimilarity (estimated at 64.28) was observed between G4 and G5 (Table 3).

In order to select promising parents to compose crossing panels in the breeding program with the aim of increase grain mineral content, the 20 best genotypes for each of the nine

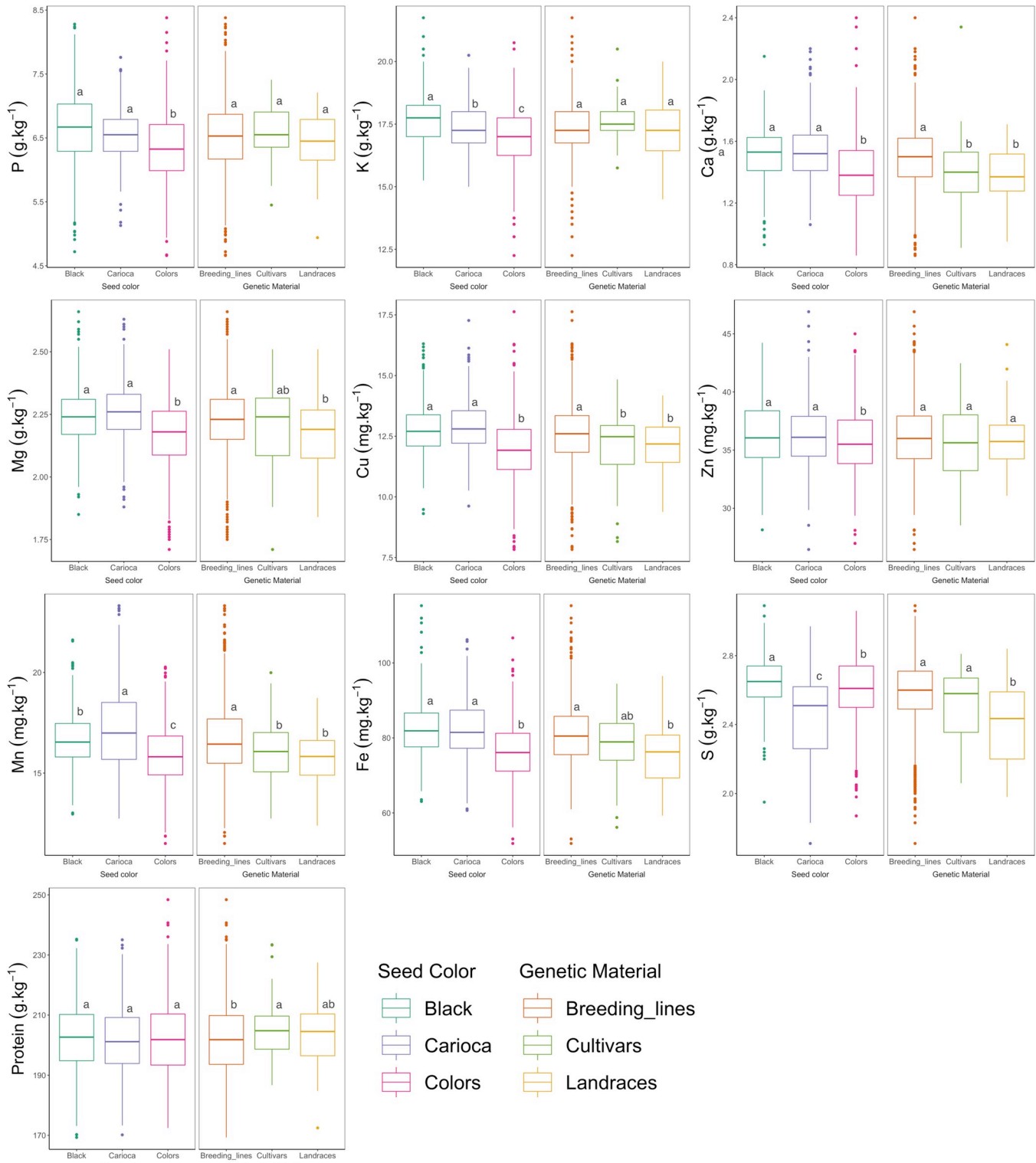

**Fig 2. Boxplot and Tukey test of grain nutrient content evaluated in 1,512 common bean accessions of the germplasm bank of the Rural Development Institute of Paraná –IAPAR–EMATER (IDR–IAPAR–EMATER), grouped according to the seed coat color and genetic material.** Groups labeled by the same lowercase letter did not differ significantly by the Tukey test ($p \leq 0.05$).

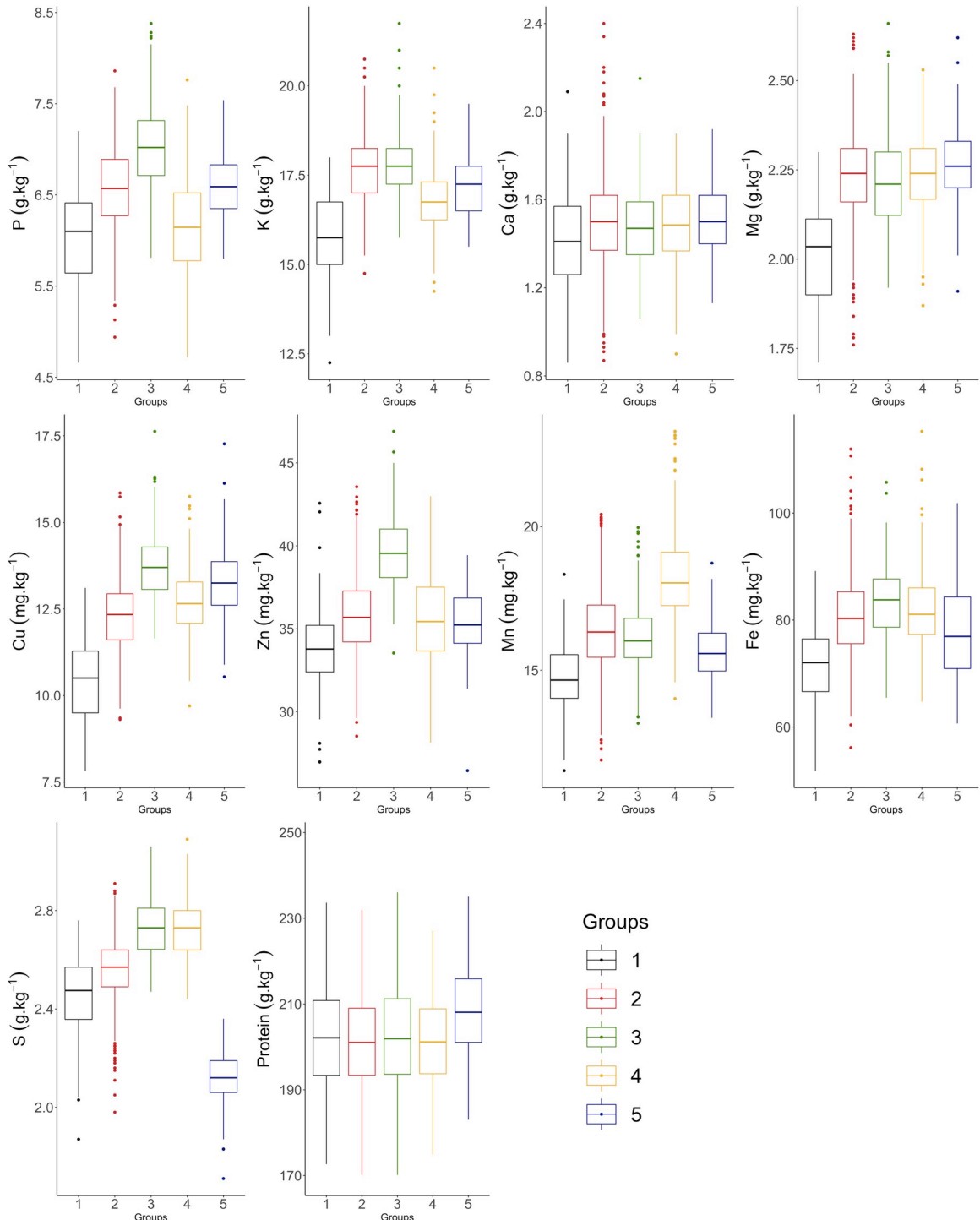

**Fig 3. Boxplot of the grain nutrient content evaluated in 1,512 common bean accessions of the germplasm bank of the Rural Development Institute of Paraná –IAPAR–EMATER (IDR–IAPAR–EMATER), separated according to the groups formed by the Ward MLM method.**

minerals and protein contents were ranked. Of all evaluated accessions, 159 were among these top 20. Of these, 127 had the highest content of only one element, 26 of two elements, three of three elements and three accessions had the highest contents of four elements.

**Table 2. Means and standard deviation of the nutrient contents in each of the five groups formed by the Ward-MLM method and contribution of the first two canonical variables of each nutrient calculated in the analysis of 1,512 common bean accessions of the germplasm bank of the Rural Development Institute of Paraná –IAPAR–EMATER (IDR–IAPAR–EMATER).**

| Nutrient | Groups | | | | | CAN | |
|---|---|---|---|---|---|---|---|
| | G1 (80) [a] | G2 (789) | G3 (202) | G4 (312) | G5 (129) | CAN1 | CAN2 |
| P (g.kg⁻¹) | 6.01 (0.51) [b] | 6.57 (0.46) | 7.03 (0.48) | 6.14 (0.51) | 6.61 (0.35) | -0.26 | 0.63 |
| K (g.kg⁻¹) | 15.68 (1.27) | 17.64 (0.93) | 17.8 (0.91) | 16.87 (0.88) | 17.23 (0.85) | -0.13 | 0.46 |
| Ca (g.kg⁻¹) | 1.4 (0.25) | 1.5 (0.21) | 1.47 (0.18) | 1.49 (0.19) | 1.51 (0.17) | -0.03 | 0.00 |
| Mg (g.kg⁻¹) | 2.01 (0.14) | 2.24 (0.12) | 2.22 (0.13) | 2.24 (0.11) | 2.25 (0.10) | -0.04 | 0.11 |
| Cu (mg.kg⁻¹) | 10.39 (1.23) | 12.31 (0.97) | 13.74 (1.00) | 12.66 (0.97) | 13.25 (1.06) | -0.08 | 0.52 |
| Zn (mg.kg⁻¹) | 33.77 (2.86) | 35.81 (2.3) | 39.72 (2.17) | 35.63 (2.81) | 35.36 (1.97) | 0.05 | 0.69 |
| Mn (mg.kg⁻¹) | 14.74 (1.32) | 16.38 (1.43) | 16.13 (1.26) | 18.26 (1.69) | 15.69 (1.09) | 0.45 | -0.16 |
| Fe (mg.kg⁻¹) | 71.43 (7.59) | 80.53 (7.61) | 83.57 (6.70) | 81.77 (7.46) | 77.8 (8.94) | 0.14 | 0.28 |
| S (g.kg⁻¹) | 2.45 (0.18) | 2.55 (0.13) | 2.73 (0.12) | 2.72 (0.12) | 2.12 (0.10) | 0.81 | 0.43 |
| Protein (g.kg⁻¹) | 201.99 (2.0) | 201.33 (1.79) | 202.52 (1.97) | 201.45 (1.74) | 208.3 (1.64) | -0.15 | -0.02 |

[a]Number of accessions;

[b] Standard deviation.

Most accessions selected among the top 20 were breeding lines, and in relation to seed coat color, 61 were of the black, 57 of the carioca, 14 of the mulatto and 6 of the white group. In an individual analysis of each nutrient, the black group accessions had the best contents for most nutrients. Accessions of the white group had the highest Ca and protein and carioca group accessions the highest Mn contents. Two accessions of the white group simultaneously had

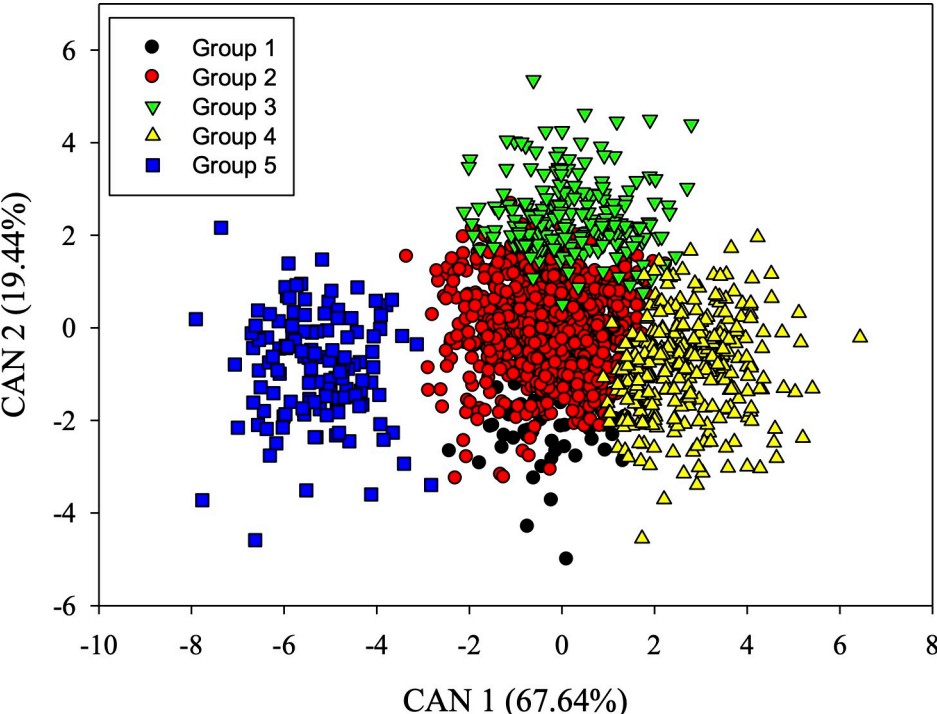

**Fig 4. Scatter plot of the first two canonical variables for the five groups formed by the Ward-MLM method.**

**Table 3. Distance between the groups formed by the Ward-MLM strategy based on the distance proposed by Matusita (1956).**

| Groups | G1 | G2 | G3 | G4 |
|--------|-------|-------|-------|-------|
| G2 | 10.73 | | | |
| G3 | 22.23 | 7.23 | | |
| G4 | 18.27 | 10.93 | 17.45 | |
| G5 | 39.77 | 28.02 | 42.53 | 64.28 |

highest Cu, Zn, S and protein contents, while two other accessions of the black group stood out with high Cu, Zn and P.

For the Fe and Zn contents, which are the main targets of biofortification in common bean, only breeding lines were among the top 20, mostly of the carioca and black groups. One line of the carioca group simultaneously had high Fe as well as Zn contents.

## Discussion

Common bean is a primary component of the diet of the population of countries in Latin America as well as in East and Southern Africa, which are regions of the world with high rates of chronic diseases associated with malnutrition [32]. Common bean is not only a crop that plays a key role in feeding populations of underdeveloped countries, but is also highly promising for breeding for biofortification. Previous studies suggest that the Fe content of common bean grain could be increased by 80% and the Zn content by up to 50% [19]. Thus, taking into account the amount of common bean consumed in these countries, the objective of biofortification of common bean for Fe would be to increase the content from 50 to 94 ppm, which would be enough to meet an adult's basic daily Fe requirement [7].

A wide variability for all mineral and protein contents was observed among accessions of the IDR–IAPAR–EMATER germplasm bank. These results indicate that it is possible to identify promising parents for exploitation in breeding programs with a view to increasing the grain nutritional contents. The existence of genetic variability and the possibility of raising the grain mineral content have been confirmed in several common bean germplasm banks in Brazil [33,34], Colombia [18,19], Portugal [35], as well as in the USA [32].

Positive correlations were observed among most nutrients, which may indicate co-segregation of the genetic factors for the different minerals [19]. This correlation is favorable for plant breeding for higher nutritional grain content, since selection for an increase in one nutrient may be favorable for other minerals simultaneously. As observed by Pinheiro et al. [35], two correlation sets (Fe-Zn-Cu-P and Ca-Mn-Mg) were formed. The correlation between Ca and Mn was also mentioned by other authors, who suggested that this correlation is due to the fact that the deposition mechanism in the grain at the uptake or transport level of the two elements is the same [32]. The correlation between P and protein can be explained by the fact that P can be accumulated by an association in the body of proteins, e.g., in the form of phytates [35]. Co-segregation evidences of Fe and Zn due to highly positive correlation was also observed in other crops, suggesting that common mechanisms regulate Fe and Zn accumulation [36].

The mean concentrations found in this study were similar to those reported by Silva et al. [33], who also evaluated mostly Brazilian genotypes of Mesoamerican origin. However, our results exceeded the contents determined in the Mesoamerican Diversity Panel studied by McClean et al. [32]. In an evaluation of common bean accessions of the Mesoamerican and Andean gene pools, Moraghan and Grafton [37] also found lower mean concentrations.

Genotypes of the Andean group as well as colored grains tend to have lower nutrient concentrations. Lower P contents in the Andean than the Mesoamerican group were described in

a study published by CIAT [18], and lower mean contents of K, Ca, Zn and Cu were observed by Ribeiro et al. [34] in an assessment of 32 common bean lines with special grain types. In this study, the mean mineral concentration of the colored group, which also includes Andean genotypes, was around 7–2% lower than that of the carioca and black genotypes.

With regard to the mineral composition of accessions of the main commercial groups black and carioca consumed in Brazil, great variability was detected within each one. However, the differences between the groups are few. Most carioca and black genotypes were grouped in G2 and G4, which were the largest groups, with greater variability and amplitude for the studied contents, while G3 was predominantly formed by black accessions and G5 by carioca accessions. In black grain genotypes, Silva et al. [33] identified higher Fe, Zn and protein contents, and higher Mg and Mn contents in carioca grain. These results partly confirm the findings of our study. In the separation of genotypes by color, the black group stood out with high K, Fe and S and the carioca group with high Zn and Mn contents.

Commonly, the main objective of common bean breeding programs is to raise yields, although the improved lines evaluated in this study performed better for all nutrients except protein. Once again, these results refute arguments that claim that modern crop breeding techniques reduce the grain quality [32]. A gradual increase of the Fe and Mg contents over time was observed, i.e., the contents increased in response to conventional breeding, less in landraces and more in cultivars and improved lines.

Only the concentration of grain crude protein was lower in the breeding lines, which can be explained by the negative correlation between grain crude protein content and yield [38]. Despite this reduction, accessions were identified with a balance between the two characteristics, enabling a concomitant improvement of both.

Due to the large number of accessions evaluated, the best 20 with the highest contents for each mineral and protein were selected. The mean of the top 20 accessions in relation to the overall mean was between 22–28% higher for Ca, Mn and Cu and around 14–18% higher for P, K, S, Mg and protein. The mean Zn and Fe contents were, respectively, 18 and 22% higher than the overall mean. Common bean is known as an excellent source of Zn and Fe, since one cup of common bean can currently supply 15 and 25%, respectively, of the recommended daily allowance for these nutrients. Still, cultivars with higher nutritional Fe and Zn contents can potentially be developed [39].

Most of these top 20 accessions were breeding lines, many of which had been improved in the IDR–IAPAR–EMATER breeding program, indicating the potential for the release of new cultivars or even including them in cross panels. In addition, many of these accessions belong to the carioca and black groups, which are the most commonly consumed grain types in Brazil. Thus, new biofortified cultivars could easily be incorporated into the diet of the Brazilian population.

In general, the variability among the evaluated accessions was high, indicating a promising potential for the development of new common bean cultivars with higher grain nutrient contents. Although not addressed as a main target of crop improvement, the nutritional grain quality has been increased indirectly by human selection, evidenced by the fact that breeding lines showed higher contents for some nutrients compared to cultivars and landraces (Fig 2). Thus, if efforts are invested in the development of biofortified cultivars, it will be possible to help combat malnutrition in countries affected by this problem.

## Supporting information

**S1 Fig. Log-likelihood graph of the ideal number of groups for 1,512 common bean accessions analyzed for grain nutrient and protein content.**
(TIF)

**S1 Table. Information of the 1,512 accessions characterized for seed nutritional content.**
1CIAT = International Center for Tropical Agriculture (Centro Internacional de Agricultura Tropical), EMBRAPA = Brazilian Agricultural Research Corporation (Empresa Brasileira de Pesquisa Agropecuária), IAC = Agronomic Institute of Campinas (Instituto Agronômico de Campinas), IDR-Paraná = Rural Development Institute of Paraná –IAPAR–EMATER (Instituto de desenvolvimento Rural do Paraná), USDA = United States Department of Agriculture, ESALQ = Universidade de São Paulo—Escola Superior de Agricultura "Luiz de Queiroz". (DOCX)

**S1 File.**
(DOCX)

**S2 File.**
(PDF)

## Author Contributions

**Conceptualization:** Jessica Delfini, Vânia Moda-Cirino, Juliana Sawada Buratto, Leandro Simões Azeredo Gonçalves.

**Data curation:** José dos Santos Neto.

**Formal analysis:** Jessica Delfini, José dos Santos Neto, Leandro Simões Azeredo Gonçalves.

**Funding acquisition:** Vânia Moda-Cirino.

**Investigation:** Juliana Sawada Buratto.

**Project administration:** Vânia Moda-Cirino.

**Supervision:** Vânia Moda-Cirino, Leandro Simões Azeredo Gonçalves.

**Writing – original draft:** Jessica Delfini.

**Writing – review & editing:** Vânia Moda-Cirino, José dos Santos Neto, Paulo Maurício Ruas, Leandro Simões Azeredo Gonçalves.

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
