## [Decision Letter · Decision Letter 0]

5 Jun 2020

PONE-D-20-08197

Diversity of nutritional content in seeds of Brazilian common bean germplasm

PLOS ONE

Dear Dr. Azeredo Gonçalves,

Thank you for submitting your manuscript to PLOS ONE. After careful consideration, we feel that it has merit but does not fully meet PLOS ONE’s publication criteria as it currently stands. Therefore, we invite you to submit a revised version of the manuscript that addresses the points raised during the review process.

We look forward to receiving your revised manuscript.

Kind regards,

Roberto Papa, PhD

Academic Editor

PLOS ONE

Journal Requirements:

"The funders had no role in study design, data collection and analysis, decision to

publish, or preparation of the manuscript."

Additional Editor Comments (if provided):

please make changes considering the comments of the reviewers and submit us the final copy of the manuscript

Reviewers' comments:

Reviewer's Responses to Questions

**Comments to the Author**

1. Is the manuscript technically sound, and do the data support the conclusions?

Reviewer #1: Yes

Reviewer #2: Yes

2. Has the statistical analysis been performed appropriately and rigorously? 

Reviewer #1: Yes

Reviewer #2: Yes

3. Have the authors made all data underlying the findings in their manuscript fully available?

Reviewer #1: Yes

Reviewer #2: No

4. Is the manuscript presented in an intelligible fashion and written in standard English?

Reviewer #1: Yes

Reviewer #2: Yes

5. Review Comments to the Author

Reviewer #1: Diversity of nutritional content in seed of Brazilian common bean germplasm

The authors presented a study on mineral nutrients (P, K, Ca, Mg, Cu, Zn, Mn, Fe and S) and protein content in a collection of 1,512 common bean accessions consisting on local, commercial and improved varieties. A total of 4 environments in 2 crop seasons were evaluated. The aim of the study is to evaluate the diversity regarding to these traits within this germplasm collection in order to identify some parental lines that can be used in future breeding programs.

The study is original and the topic is interesting since the biofortification is one the target for breeding programs nowadays. Nevertheless, some points specially regarding to the material and methods need to be clarify in order to understand better the study.

Abstract:

The abstract is well written and summarizes the information included in the work.

Introduction:

In the introduction, the authors presented the problematic and the objective of the study is clearly established.

Material and Methods:

Line 109: different center of origin…Describe how many of the 1512 accessions are Mesoamerican and how many Andean since this data is later used in the results and it is useful information for the reader.

Line 138: the factor 6.25 is used to convert total seed N to protein content. Reference this conversion factor for common bean.

According to I.E. Ezeagu et al. / Food Chemistry 78 (2002) 105–109. Crude protein analysis conventionally based on multiplying the total nitrogen (N) (Kjeldalh N) by a factor of 6.25 has been suspect. Different conversion factors recommended for cereals and grain products range between 5.70 and 5.83; for dry grain legumes 5.46–5.71 is recommended; while 5.18–5.46 is for nuts and seeds (FAO, 1982).

Considered this reference. François Mariotti, Daniel Tomé, Philippe Mirand. Converting Nitrogen into Protein – Beyond 6.25 and Jones’ Factors. Critical Reviews in Food Science and Nutrition, Taylor & Francis, 2008, 48 (2), pp.177-184. ff10.1080/10408390701279749ff. ffhal-02105858f

Please discuss this and reference the conversion factor used in this study for common bean.

Line 145 -146: the authors said: with these data, a Tukey test and boxplot were performed to show the distribution of the accessions grouped by the criteria using R packages….

Some points are needed to be clarify in this sentence in order to explain better the methods.

Please specify what data is used for the contrast test in order to follow better the methods…. the mean values for each analyzed trait per accession, per seed color group or per genetic material.

Regarding the Tukey test is this multiple means comparison test supporting another global mean test (ANOVA or Krukall wallis etc)?

Specify also in the text the alpha level for the statistical test.

Line 148: Ward-MLM. Specify the complete name of these test. Ward- Modified Location Model.

Lines 159-160: This sentence is a bit confusing; the authors presented a correlation network made with qgraph, but which data is correlated? the dissimilarity coefficients, the distance coefficients, the mean values. Please specify in order to specify this issue.

It would be also interesting to specify what method do you use for correlating the data with qgraph, (cor_auto?)

Results:

Line 182: Positive and median correlations, what do you mean with median correlations?. Are all these correlations significant? Please include in the results the signification level of the correlations to better interpretation the results.

Lines 190-192: This sentence is a bit confusing please rewrite.

Line 197: add (Figure 2)

Line 227-241: In this section of the results the composition of the groups obtained by Ward-MLM method is describe, but the table 2 just reflects the total number of accessions of each group but not the characteristics of the accessions included in each one. It might be interesting to include this information the supplementary material since it is presented as results.

Line 243: It might be Figure 5 instead of Table 2

Line 247: table 2 instead of Fig 5

Lines 258- 263: this paragraph is not easy to follow. Please re-write it.

Discussion:

In general, the discussion is well written and it adresses the main points of the results.

line 336: proper bibliography format [36]

Tables and figures:

Table 1: please review the legend of the table, the superindex a and the cross might be changed.

Figure 1: correlation network, please include the significance level for each correlation or at least for those that are significant.

Figure 2: include significant level considering for the tukey test (plabeled with different letters in the plot indicate differences between group means?

Figure 3. Might be interesting to include the plots for pseudo-t2 and pseudo-F in the supplementary materials as it is the criteria used for the grouping. Suplementary material.

Table 2: in the legend (line 225) delete CV

In general, for all the figures, improve printing quality for seeing details.

The bibliography is well formated

Reviewer #2: In general the study is interesting: the Abstract provide a first insight into the results obtained, giving importance to commercial varieties widely diffused in Brazil as carioca and black beans, arising major interest for partecipatory breeding and breeding strategies in the next future. The Introduction is complete and well structured with references righlty reported. Materials and Method are well organized as the workflow strategy is highly reccomended for these type of experiment and confirmed by previous studies applying the same methodology. Results are clear as they follow the order of the analysis conducted as explained in materials and methods.

In general the study is well conducted and explained but maybe some slight adjustments could be done. I’ve resumed some points below (with reference to manuscript lines).

Line 35 – 39: not clear the distinction among gene pools… I would specify in line 35 that carioca and black market group is characterized by mesoamerican accessions.

Line 91 : maybe some very fast highlights on the conclusions reached by these two authors (very fast because you already explain this in the conclusions, but could be useful to touch the topic in the introduction)… for example: “the mesoamerican gene pool gave higher lectin, clacium,phosphorus,sulfur and zinc than the andean but lower phaseolin and iron” or “The major gene pools differed significantly in almost all grain constituents… (Discussion of Islam, 2002), or see Cominelli et al., 2019.

Line 106: In material and method: will you provide a table with the list of the accessions with the relative gene pool, seed color and genetic material specified? (this is the MAJOR IMPORTANT POINT!!!!)

Line 142: maybe would be nice to make a simple PCA to see how accessions behave considering variables that you use to create your preliminary groups. I would add also the belonging to a gene pool that at the end gives also a correlation with the dimensions of the seeds

Line 146: Tukey test and boxplot could be done also separating by gene pools?

Line 220: In Table 2, you show the different groups with the relative contribution of the first two canonical variables of each nutrient and means/st.dev of the nutrient contents in each group. However, while describing the groups (just below the table) you specify the descriptors of each group but without a rational order. Eventually, would be nice to have descriptors cited always in the same order for all the groups and I would give a world to each category you considered at the beginning, thus: seed color, genetic material and I would add also the belonging to the gene pool. (Eventually a pCoA for each group would be useful to see how accessions in each group distribute in relation to the categories mentioned above?)

Line 304: maybe a fast citations of the co-segregation evidences of Fe and Zn too?

Line 311: rightly you talk about gene pools but in your experiment seems that you skip the belonging to a gene pool as an important factor. Maybe it has a sense considering your experiment but it is not clear to the reader… this should be specified somewhere as I’ve showed in the comments above.

Line 356: a reference when you say “increased indirectly by human selection”.

6. PLOS authors have the option to publish the peer review history of their article (what does this mean?). If published, this will include your full peer review and any attached files.

Reviewer #1: Yes: Ester Murube Torcida

Reviewer #2: No

---

## [Author Response · Author response to Decision Letter 0]

27 Jul 2020

Dear editor,

On behalf of my collaborators, I am submitting the manuscript titled “Diversity of nutritional content in seeds of Brazilian common bean germplasm”, after reviewing as requested by referees.

We believe that we addressed all questions and concerns and we are grateful for the opportunity to improve our manuscript. All changes are marked in the text as requested and specific answers are presented as follows:

Response to reviewers 

Reviewer #1: 

1. Line 109: different center of origin…Describe how many of the 1512 accessions are Mesoamerican and how many Andean since this data is later used in the results and it is useful information for the reader.

Response: The information was added to the main text.

2. Line 138: the factor 6.25 is used to convert total seed N to protein content. Reference this conversion factor for common bean.

According to I.E. Ezeagu et al. / Food Chemistry 78 (2002) 105–109. Crude protein analysis conventionally based on multiplying the total nitrogen (N) (Kjeldalh N) by a factor of 6.25 has been suspect. Different conversion factors recommended for cereals and grain products range between 5.70 and 5.83; for dry grain legumes 5.46–5.71 is recommended; while 5.18–5.46 is for nuts and seeds (FAO, 1982).

Considered this reference. François Mariotti, Daniel Tomé, Philippe Mirand. Converting Nitrogen into Protein – Beyond 6.25 and Jones’ Factors. Critical Reviews in Food Science and Nutrition, Taylor & Francis, 2008, 48 (2), pp.177-184. ff10.1080/10408390701279749ff. Please discuss this and reference the conversion factor used in this study for common bean.

Response:

We understand and appreciate your concern about that. This is a controversy question and some crossed opinions can be found about this matter over all types of food. Usually for common beans the 6.25 factor, recommended by the Association of Official Analytical Chemists (AOAC) (now referenced in the text) is used. Therefore, as the purpose of this article is to characterize the germplasm bank and also to compare the values obtained with previous works, we chose to use the 6.25 factor. We understand that the proportion of protein among the accessions will not be affected by the multiplying factor, which would not affect the diversity study.

Reference added to the text:

Association of Official Analytical Chemists - International [AOAC]. Official methods of analysis. 18th ed. Gaithersburg: AOAC; 2005. 

Some references with articles about the bean crop:

Silva CA, Abreu Â de FB, Ramalho MAP, Maia LGS. Chemical composition as related to seed color of common bean. Crop Breed Appl Biotechnol. 2012;12: 132–137. 

Wiesinger JA, Cichy KA, Glahn RP, Grusak MA, Brick MA, Thompson HJ, et al. Demonstrating a Nutritional Advantage to the Fast-Cooking Dry Bean (Phaseolus vulgaris L.). J Agric Food Chem. 2016;64: 8592–8603. doi:10.1021/acs.jafc.6b03100

Celmeli T, Sari H, Canci H, Sari D, Adak A, Eker T, et al. The nutritional content of common bean (phaseolus vulgaris l.) landraces in comparison to modern varieties. Agronomy. 2018;8. doi:10.3390/agronomy8090166

Coelho CMM, Bellato C de M, Santos JCP, Ortega EMM, Tsai SM. Effect of phytate and storage conditions on the development of the ‘ hard-to-cook .’ J Sci Food Agric. 2007;1243: 1237–1243. doi:10.1002/jsfa

Jannat S, Shah AH, Shah KN, Kabir S, Ghafoor A. Genetic and nutritional profiling of common bean (Phaseolus vulgaris l) germplasm from Azad Jammu and Kashmir and exotic accessions. J Anim Plant Sci. 2019;29: 205–214.

3. Line 145 -146: the authors said: with these data, a Tukey test and boxplot were performed to show the distribution of the accessions grouped by the criteria using R packages….

Some points are needed to be clarify in this sentence in order to explain better the methods. Please specify what data is used for the contrast test in order to follow better the methods…. the mean values for each analyzed trait per accession, per seed color group or per genetic material. Regarding the Tukey test is this multiple means comparison test supporting another global mean test (ANOVA or Krukall wallis etc)? Specify also in the text the alpha level for the statistical test.

Response: The complete description of the analytic analyses was added to the text.

“The accessions were separated based on two criteria: seed coat color (black, carioca and colored) and genetic material (cultivars, landraces and breeding lines). For these two criteria an analysis of variance (ANOVA) followed by a Tukey test (p < 0.05) were performed for each case to compare significantly differences between the formed groups. To visualize the distribution of the accessions grouped by each criteria a boxplot graph was made.”

4. Line 148: Ward-MLM. Specify the complete name of these test. Ward- Modified Location Model.

Response: Done

5. Lines 159-160: This sentence is a bit confusing; the authors presented a correlation network made with qgraph, but which data is correlated? the dissimilarity coefficients, the distance coefficients, the mean values. Please specify in order to specify this issue.

It would be also interesting to specify what method do you use for correlating the data with qgraph, (cor_auto?)

Response: The sentence was moved to a better place that makes more sense in relation to the order of the analyzes shown in the results and in the MeM, and more details was added. 

6. Line 182: Positive and median correlations, what do you mean with median correlations? Are all these correlations significant? Please include in the results the signification level of the correlations to better interpretation the results.

Response: Only edges of significant correlations are shown in the graph (p ≤ 0.05). Added this information to the figure legend.

7. Lines 190-192: This sentence is a bit confusing please rewrite.

Response: Done

8. Line 197: add (Figure 2)

Response: Done

9. Line 227-241: In this section of the results the composition of the groups obtained by Ward-MLM method is describe, but the table 2 just reflects the total number of accessions of each group but not the characteristics of the accessions included in each one. It might be interesting to include this information the supplementary material since it is presented as results.

Response: The table 2 shows the number of accessions, means and standard deviation of the nutrients contents in each group. Also, a supplementary table are now included showing the access belonging to each Ward-MLM group.

10. Line 243: It might be Figure 5 instead of Table 2

Response: Done

11. Line 247: table 2 instead of Fig 5

Response: Done

12. Lines 258- 263: this paragraph is not easy to follow. Please re-write it.

Response: Done

13. line 336: proper bibliography format [36]

Response: Done

14. Table 1: please review the legend of the table, the superindex a and the cross might be changed.

Response: Done

15. Figure 1: correlation network, please include the significance level for each correlation or at least for those that are significant.

Response: Done

16. Figure 2: include significant level considering for the tukey test (plabeled with different letters in the plot indicate differences between group means?

Response: Done

17. Figure 3. Might be interesting to include the plots for pseudo-t2 and pseudo-F in the supplementary materials as it is the criteria used for the grouping. Suplementary material.

Response: Done

18. Table 2: in the legend (line 225) delete CV

Response: Done

19. In general, for all the figures, improve printing quality for seeing details.

Response: All figures were submitted to PACE and fulfilled the required quality parameters. 

Reviewer #2: 

1. Line 35 – 39: not clear the distinction among gene pools… I would specify in line 35 that carioca and black market group is characterized by Mesoamerican accessions.

Response: Done

2. Line 91 : maybe some very fast highlights on the conclusions reached by these two authors (very fast because you already explain this in the conclusions, but could be useful to touch the topic in the introduction)… for example: “the mesoamerican gene pool gave higher lectin, clacium,phosphorus,sulfur and zinc than the andean but lower phaseolin and iron” or “The major gene pools differed significantly in almost all grain constituents… (Discussion of Islam, 2002), or see Cominelli et al., 2019.

Response: The information was added to the introduction.

3. Line 106: In material and method: will you provide a table with the list of the accessions with the relative gene pool, seed color and genetic material specified? (this is the MAJOR IMPORTANT POINT!!!!)

Response: A supplementary table with the list of accession with center of origin, seed coat color and genetic material is being provided now.

4. Line 142: maybe would be nice to make a simple PCA to see how accessions behave considering variables that you use to create your preliminary groups. I would add also the belonging to a gene pool that at the end gives also a correlation with the dimensions of the seeds

Response: A PCA was performed as suggested, however there was no different behavior for each group considering the preliminary groups. Regarding the gene pool, it would not be possible to do this due to the low number of accessions of Andean origin.

5. Line 146: Tukey test and boxplot could be done also separating by gene pools?

Response: The number of accessions of Andean gene pool is very low compared to the Mesoamerican, and since all of the Andean accessions have colored grains and showed similar results to the colored Mesoamerican accessions they were grouped together in the analyses.

6. Line 220: In Table 2, you show the different groups with the relative contribution of the first two canonical variables of each nutrient and means/st.dev of the nutrient contents in each group. However, while describing the groups (just below the table) you specify the descriptors of each group but without a rational order. Eventually, would be nice to have descriptors cited always in the same order for all the groups and I would give a world to each category you considered at the beginning, thus: seed color, genetic material and I would add also the belonging to the gene pool. (Eventually a pCoA for each group would be useful to see how accessions in each group distribute in relation to the categories mentioned above?)

Response: Looking at the characteristics of the accessions that formed each group, the color of the coat was the one that most contributed to the formation of the groups, so this characteristic being the most discussed. In the case of origin, only G1 comprised a significant number of Andean accessions, so origin was only mentioned for this group. Genetic material was not mentioned in the discussion of the groups because, as the dataset is formed mostly by breeding lines, in all groups the breeding lines were the majority and in none of the cases this characteristic becomes relevant in the formation of the groups. 

We appreciate your suggestion to do a PCoA for each of the two categories (seed color and genetic material) but it would generate too much graphs and would difficult the correct interpretation of the graphs. Knowing that genetic material didn’t contributed to the formation of the groups we opted for a more general view of the groups. Now a supplementary table was provided with the information of access belonging to each Ward-MLM group, and also, the distribution of each group related to the nutrients composition are shown in the boxplots (Fig 3). 

7. Line 304: maybe a fast citations of the co-segregation evidences of Fe and Zn too?

Response: Done 

8. Line 311: rightly you talk about gene pools but in your experiment seems that you skip the belonging to a gene pool as an important factor. Maybe it has a sense considering your experiment but it is not clear to the reader… this should be specified somewhere as I’ve showed in the comments above.

Response: As cited before, the number of accessions Andean gene pool is very low compared to the Mesoamerican, that’s why studies were not performed exclusively for origin. The number of Andean accessions was added to MeM to evidence the low number of those in the study

9. Line 356: a reference when you say “increased indirectly by human selection”.

Response: Done

---

## [Decision Letter · Decision Letter 1]

3 Sep 2020

Diversity of nutritional content in seeds of Brazilian common bean germplasm

PONE-D-20-08197R1

Dear Dr. Azeredo Gonçalves,

We’re pleased to inform you that your manuscript has been judged scientifically suitable for publication and will be formally accepted for publication once it meets all outstanding technical requirements.

Kind regards,

Roberto Papa, PhD

Academic Editor

PLOS ONE

Additional Editor Comments (optional):

Reviewers' comments:

Reviewer's Responses to Questions

**Comments to the Author**

1. If the authors have adequately addressed your comments raised in a previous round of review and you feel that this manuscript is now acceptable for publication, you may indicate that here to bypass the “Comments to the Author” section, enter your conflict of interest statement in the “Confidential to Editor” section, and submit your "Accept" recommendation.

Reviewer #1: All comments have been addressed

Reviewer #2: All comments have been addressed

2. Is the manuscript technically sound, and do the data support the conclusions?

Reviewer #1: Yes

Reviewer #2: Yes

3. Has the statistical analysis been performed appropriately and rigorously? 

Reviewer #1: Yes

Reviewer #2: Yes

4. Have the authors made all data underlying the findings in their manuscript fully available?

Reviewer #1: Yes

Reviewer #2: Yes

5. Is the manuscript presented in an intelligible fashion and written in standard English?

Reviewer #1: Yes

Reviewer #2: Yes

6. Review Comments to the Author

Reviewer #1: After considering all the changes suggested by the reviewers, I think the manuscript improves in terms of clarity and ease of understanding the study. As I already included in my first review, I consider that the printing quality of the images is not very good and I urge the journal editing team to check this issue in the final publication.

It seems to be a very interesting study that highlight the importance of studying nutritional characteristics in a crop as important as the common bean.

Reviewer #2: Authors have efficiently addressed the doubts arose by the reviewers in an exhaustive way. Tables and figures have been updated as suggested. The new submitted document is more legible and understandable by readers, in particular for the part relative to matherials and methods. Authors gave complete explanations to reviewers on decisions and modifications made after the first review. The article is now complete and ready for submission.

7. PLOS authors have the option to publish the peer review history of their article (what does this mean?). If published, this will include your full peer review and any attached files.

Reviewer #1: **Yes: **Ester M. Murube Torcida

Reviewer #2: No

---

## [Editor Report · Acceptance letter]

15 Sep 2020

PONE-D-20-08197R1 

Diversity of nutritional content in seeds of Brazilian common bean germplasm 

Dear Dr. Azeredo Gonçalves:

I'm pleased to inform you that your manuscript has been deemed suitable for publication in PLOS ONE. Congratulations! Your manuscript is now with our production department. 

Kind regards, 

on behalf of

Prof. Roberto Papa 

Academic Editor

PLOS ONE